# Expression of transgenic biotin ligases in inducible neuronal murine cell lines by integration into the mHipp11 gene locus

**Lisa Feicht, Aaron Dangel**☉¤, **Ralf-Peter Jansen**☉*

Interfaculty Institute of Biochemistry (IFIB), University of Tübingen, Germany

¤ Current address: Division Translational Genomics of Neurodegenerative Diseases, Hertie-Institute for Clinical Brain Research, Tübingen, Germany
* ralf.jansen@uni-tuebingen.de

## Abstract

Biotin proximity labeling is a powerful method for identifying proteins associated with a specific organelle, a bait protein, or RNA. It requires the expression of a modified biotin ligase by transient transfection or from a stably integrated expression construct. Because such stable integration of transgenes into stem cells can lead to silencing during differentiation, targeting a biotin ligase to a genomic safe harbor site would be beneficial. Here, we report on the successful targeting and expression of two biotin ligase constructs to the mouse Hipp11 locus during neuronal differentiation. While randomly integrated MicroID and TurboID are expressed and active in mouse embryonic stem cells (mESCs), expression ceases upon differentiation into mESC-derived neurons, which is independent of the promoter used. In contrast, targeting of the same expression cassette to the mHipp11 locus results in expression, correct localization, and biotinylation activity not only in mESCs but also in neurons 8–10 days after differentiation. This demonstrates that the mouse Hipp11 locus is a promising genomic integration site for transgenic biotin ligases in mESCs and mESC-derived neurons.

## Introduction

Transgene expression in mammalian cells has become an important and powerful tool for studying the molecular and cellular function of a gene or protein of interest. Common approaches to this end include transient or stable transfection using temporarily maintained plasmids or genomically integrated DNA, respectively. As each method has advantages and disadvantages, the ideal approach depends on the cell type and experimental setup (see [1,2] for an overview). Ideally, transgene transfection and expression should have high transfection efficiency and reproducibility, low cell toxicity, minimal impact on cell physiology, and ease of use. The generation of stable cell lines expressing a gene of interest provides an advantage over transient gene expression by reducing variations in transfection efficiency and maintaining expression for long-term studies. Typically, stable cell line generation involves the transfection of a plasmid containing a promoter that drives the expression of the gene of interest and an antibiotic marker to select cells that have integrated the foreign DNA at a random location

**Data availability statement:** All relevant data are within the manuscript and its Supporting Information files.

**Funding:** RPJ was funded by a grant of the Deutsche Forschungsgemeinschaft (grant no. DFG JA696/10-2) in the course of the DFG-funded Research Unit FOR2333 (https://gepris.dfg.de/gepris/projekt/270067186). LF was supported by the International Max Planck Research School "From Molecules to Organisms" (https://www.phd.tuebingen.mpg.de/imprs). The funders had no role in study design, data collection and analysis, decision to publish, or preparation of the manuscript.

**Competing interests:** The authors have declared that no competing interests exist.

in the genome. However, stable gene integration at random sites can result in unpredictable expression levels, tend to be silenced over time, and/ or may even disrupt the expression of endogenous genes near the integration site [3–5]. In addition, transgene expression can be strongly influenced by the genomic region, the state of the surrounding chromatin at the integration site, and the promoter used to express the transgene [5]. To overcome these problems, the transgene of choice can be targeted to a 'safe-harbor locus' [6]. These are intragenic or intergenic regions present in the mammalian genome that allow stable expression of integrated transgenes without affecting the host cell. Typical loci used in the mouse genome are *Rosa26* [7–9] and *Hprt1* [10–12]. Alternative loci are intergenic regions that do not contain endogenous promoter elements. An example for this is the mouse Hipp11 intergenic region located on chromosome 11 and situated between two ubiquitously expressed and oppositely transcribed genes, *Eif4enif1* and *Drg1*. This genomic region has been previously characterized as a safe-harbor locus in mice, pigs and humans [13–16]. We evaluated the mHipp11 locus as a potential integration site for proximity labeling enzymes [17], since expression cassettes of these proteins are often integrated into the genome of host cells. Proximity labeling is a relatively novel method for biotin tagging of endogenous interacting partners, allowing their isolation and identification. It relies on the expression of promiscuous enzymes (modified biotin ligases or peroxidases), often as fusion proteins to generate a highly reactive and diffusible biotin species for protein or RNA labeling [17,18]. The mHipp11 locus as a genomic targeting site for this type of enzymes could be particularly useful for cell lines or organisms whose commonly used safe-harbor sites are occupied by other transgenes, e.g., for inducible systems such as the Cre-lox system or the doxycycline regulatory cassette [19,20]. One such cell line is the genetically modified mouse embryonic stem cell (mESC) line that can be differentiated into neurons via doxycycline-inducible expression of the neurogenic transcription factor Achaete-scute homolog 1 (ASCL1) [19,21,22]. In this line, the components for the doxycycline-inducible ASCL1 expression have been integrated into the Rosa26 and Hprt1 safe-harbor loci [23]. Here, we describe the integration and functional expression of a biotin ligase ('MicroID') at the Hipp11 intergenic region using a CRISPR/Cas9 approach and demonstrate its superiority over standard random integration in mESCs and mESC-derived neurons.

## Methods

### Plasmid constructs

The mHipp11 knock-in vector was kindly provided by Vincent Kelly (Trinity College Dublin, Ireland). The 2xMCP-GFP-MicroID protein was expressed either from a human cytomegalovirus (CMV) or mouse phosphoglycerokinase (mPGK) promoter. To generate the corresponding plasmids used for targeting, a DNA fragment containing the coding region of the MicroID biotin ligase [24] was amplified with a proofreading polymerase from plasmid pSF3-MicroID ([24]; gift from Julian Bethune, Hamburg University of Applied Science) using primers #7040 and #7041 (S1 Table). The resulting PCR product and plasmid pcDNA3.1(+) CMV-NLS-2XMCP-eGFP-BirA* (unpublished) were digested with NotI and Bsp1407 to release BirA* and subsequently vector and PCR product were used for ligation to generate pcDNA3.1(+) CMV-NLS-2XMCP-eGFP-MicroID. To generate the repair template pBSSKII-C57BL6-J-Hipp11-CMV-MicroID, we used primer #7378 and #7379 to amplify the CMV-NLS-2XMCP-eGFP-MicroID insert from the above-mentioned plasmid. PCR fragments were cloned into a SpeI-digested pBSSKII-C57BL6-J-Hipp11 knock-in vector using the SLIC method [25].

To generate the pBSSKII-C57BL6-J-Hipp11-mPGK-MicroID repair template, we used plasmid pUC57-HDR-Arc to amplify the mPGK promoter with a proofreading polymerase and primers #7349 and #7353. The resulting PCR fragment and the pcDNA3.1(+)

CMV-NLS-2XMCP-eGFP-MicroID plasmid were digested with MluI and AflII and ligated to obtain pcDNA3.1(+) mPGK-NLS-2XMCP-eGFP-MicroID. The sequence containing mPGK-NLS-2xMCP-eGFP-MicroID construct was amplified from the plasmid pcDNA3.1(+) mPGK-NLS-2XMCP-eGFP-MicroID with primers #7377 and #7378 and the product cloned into the SpeI-digested pBSSKII-C57BL6-J-Hipp11mHipp11 knock-in vector using SLIC [25]. pBSSKII-C57BL6-J-Hipp11-CMV-MicroID and pBSSKII-C57BL6-J-Hipp11-mPGK-MicroID plasmids were verified by sequencing.

### Ethics statement

The project did not involve work with animals or human subjects.

### Cell culture and differentiation into mESC-derived neurons

mESCs were cultured as previously described in [22] with the following modification: mESCs were split twice a week 1:25–1:40 and plated on 0.1% gelatin-coated flasks. For generation of mESC-derived neurons $1.5\text{x}10^6$ cells were plated in 12 mL AK medium [22] per 75 cm$^2$ suspension flask.

### Stable cell line generation: CRISPR-Cas9 mediated knock-in of MicroID cassettes at the mHipp11 locus

To prepare the ribonucleoprotein particle (RNP) complex 1 μL Cas9 (61 μM), 1,4 μL (100 μM) gRNA and 0.6 μL 1x PBS were mixed and incubated for 30 min at room temperature. For nucleofection, the manufacturer's instructions for P3 Primary Cell 4D-Nucleofector X Kit were followed. Briefly, $\sim 6\text{x}10^6$ cells were resuspended in 90 μL Nucleofection solution and mixed thoroughly with 4 μL of RNP complex and 6.6 μL repair template (1.5 μg/μL; 10 μg plasmid in total). Post nucleofection, 600 μL 80/20 medium was added to cuvettes and $\sim 300$ μL of cell suspension was transferred to one 10 cm$^2$ dish. After three days (72 h) post nucleofection, cells were cultured with 175 μg/mL hygromycin for a total of seven days for antibiotic selection. To generate monoclonal cell lines, 3–4 mL of cell suspension with a concentration of $2.5$–$3.5\text{x}10^6$ cells/mL were thoroughly dissociated to single cells in FACS buffer (1x PBS supplemented with 2% FBS and 0.5 mM EDTA, filtered and stored at 4°C) and filtered through a cell strainer. Single cells were sorted based on eGFP expression into 0.1% gelatin-coated 96-well plates at the FACS Core Facility of the University Hospital Tübingen and expanded.

### Isolation of genomic DNA and transgenic screening

eGFP-positive clones were grown in T25 cm$^2$–T75 cm$^2$ flasks, harvested and lysed in 1 mL of gDNA lysis buffer (100 mM NaCl, 10 mM Tris-HCl pH 8.0, 25 mM EDTA pH 8.0, 0.5% SDS (w/v), 500 μg/mL Proteinase K) at 55°C overnight. After incubation, 300 μL of phenol/chloroform/isoamyl alcohol was added and rotated at room temperature for 60 min before centrifugation (10.000 rpm, 5 min). The upper phase was mixed with 1/10 volume of 3 M NaAc pH 5.2 and three volumes of 96% EtOH. The gDNA was pelleted by centrifugation (10 min, 10.000 rpm, 4°C) and dried at room temperature for 10 min. DNA was mixed with 100–500 μL 1x TE buffer and incubated at 55°C for $\sim 4$ h to resuspend. The concentration was determined by nanodrop measurements. For screening of successful integration, we used 100 ng/μL as gDNA template with the following primer pairs: #7388 + #7399 (mPGK-MicroID clones) or #7388 + #7418 (CMV-MicroID clones) for the left homology arm (LHA) junction, #7398 + #7389 for the right homology arm (RHA) junction, #7413 + #7414 for eGFP-MicroID integration and #7390 + #7391 for checking hetero-/homozygous integration.

## Biotin labeling by MicroID biotin ligase in mESCs and mESC-derived neurons

For both cell types, cells were treated with 50 μM biotin (DMSO as negative control) and incubated at 37°C for different time points as indicated. All samples were placed on ice and washed five times with ice-cold 1x PBS to remove excess biotin. The cells were directly lysed with lysis buffer (50 mM Tris-HCl, pH 7.4, 150 mM NaCl, 2.5 mM MgCl$_2$, 1 mM DTT, 1% Triton-X-100, 1x cOmplete™ EDTA-free Protease Inhibitor Cocktail (Sigma-Aldrich), scraped from the surface and incubated for 10 min on ice. Lysates were cleared by centrifugation at 13,200 g for 15 min at 4°C. Protein concentration was quantified using a Bradford assay. Lysates were stored at −80°C until further use.

## Western blotting and streptavidin pulldown

For all western blots, 20 μg–25 μg of protein was separated on SDS-PAGE gels, transferred to nitrocellulose membranes, and then stained by Ponceau S (10 min in 0.1% (w/v) Ponceau S in 5% acetic acid/water). The blots were blocked in 5% (w/v) milk in 1x TBST (Tris-buffered saline, 0.1% Tween 20) for at least 30 min at room temperature. Blots were incubated with primary antibodies in 3% milk (w/v) in 1x TBST overnight at 4°C, washed three times with 1x TBST for 10 min each, then incubated with secondary antibodies in 3% milk (w/v) in 1x TBST for 1 h at room temperature. For the streptavidin-HRP conjugate, the blots were blocked in 3% BSA in 1x TBST overnight at 4°C, then incubated with 0.5 μg/mL streptavidin-HRP conjugate for 1 h at room temperature. The blots were washed three times with 1x TBST for 10 min each before development with Pierce™ ECL Western Blotting substrate (Thermo Fisher). Images were acquired on a ChemiDoc Imaging System (Bio-Rad). Upon detection of over-saturated signal (indicated by a red signal), for simplification the image was processed and converted to grayscale. For pulldown of biotinylated proteins before western blotting we used 250 μg protein lysate for mESCs and 500 μg for mESC-derived neurons. Lysates were mixed with 20 μL streptavidin-coupled magnetic beads (GE Healthcare) and incubated for 70 min at room temperature. After washing with lysis buffer, 1 M KCl, 0.1 M sodium carbonate buffer, 2 M urea in 10 mM Tris-HCl (pH 8), and another round of IP lysis buffer (each wash for 1 min at room temperature), proteins were eluted in 3x sample buffer supplemented with 2 mM biotin and 20 mM DTT (10 min at 95°C).

## Immunofluorescence and live-cell imaging

For microscopy experiments (immunofluorescence and live-cell imaging), 2.5x–7.5x 10$^4$ cells were seeded on glass slides or live-cell imaging chambers coated with either 0.1% gelatin-coated slides (for mESCs) or 0.01% poly-L-ornithine plus 5 μg/μL laminin (for neurons) and grown overnight in case of mESCs or cultured as described above in the section "Cell culture and differentiation into mESC-derived neurons". For live-cell imaging, cells were washed once with pre-warmed 1x PBS and imaged in live-cell imaging buffer ([26]; HEPES-buffered solution (HBS) containing 20 mM HEPES pH 7.4, 119 mM NaCl, 5 mM KCl, 2 mM CaCl$_2$, 2 mM MgCl$_2$, 30 mM glucose). For immunofluorescence experiments we followed the procedure described in [22] with following modifications: cells were washed twice with 1x PBS and fixed with 3.7% (v/v) paraformaldehyde for 20 min at room temperature. After two washing steps with 1x PBS, cells were permeabilized with 0.2% Triton-X-100 for 15 min at room temperature. Cells were again washed three times with 1x PBS and blocked with 10% BSA (w/v) for at least 2 h at room temperature or overnight at 4°C. Primary antibodies (S2 Table) were diluted in 3% (w/v) BSA and incubated overnight at 4°C. Afterwards, cells were washed three times with 1x TBST for 10 min. Secondary antibodies (S2 Table) and the streptavidin-AF594

or AF647 conjugate (Thermo-Fisher) were diluted in 3% BSA and cells were incubated for 1 h at room temperature. The coverslips were washed twice with 3% (w/v) BSA and once with 1x TBST for 10 min. Cells were counterstained with 0.2 µg/mL DAPI for 10 min, followed by three washing steps with 1x PBS before mounting with Vectashield Vibrance Antifade Mounting Medium (Biozol). Cells were imaged with a Zeiss CellObserver equipped with a Colibri LED illumination unit. For live cell imaging experiments, we imaged cells with 30–80% LED intensity and 200–300 ms exposure time (S2 Fig, S4 Fig). For AF594-streptavidin immunofluorescence we imaged cells with 25% LED intensity and 150 ms exposure time or 50% LED and 150 ms exposure time (S4 Fig). Immunofluorescene imaging of mESCs (anti-HA and AF647-streptavidin) was performed with 70% LED intensity and 250 ms exposure for HA and 20% intensity and 150 ms for AF647-streptavidin (Fig 2A). For imaging of neurons (Fig 3) with anti-GFP and AF647-streptavidin we used 50% LED intensity and 150 ms exposure time for both channels. Detection of Oct3/4, NF200, and MAP2 was done with 30% LED intensity and 150 ms exposure time, 20% intensity and 150 ms, or 75% intensity and 150 ms, respectively (S3 Fig).

### RNA isolation and RT-qPCR

RNA was isolated with TRI Reagent (Sigma) according to the manufacturer's protocol. For DNAse treatment, 320–1000 ng total RNA was digested with RQ1 RNAse free DNase (Promega). DNAse-treated RNA was subsequently used for cDNA synthesis using High-Capacity cDNA Reverse Transcription Kit (Thermo Fisher). For qPCR using a Fast Sybr Green Master Mix (Applied Biosystems), cDNA was diluted 1:15 or 1:25, and mixed with primer pair #7557 + #7558 for eGFP-MicroID or #7137 + #7138 for GAPDH detection. Statistical significance for MicroID expression in neurons was analyzed using an unpaired Student's t test.

## Results

### Generation and validation of mHipp11 CMV MicroID and mHipp11 mPGK MicroID cell lines

While establishing the expression of biotin ligases for RNA proximity labeling [27,28] in mESCs and mESC-derived neurons ('iNeurons'; [21]), we realized that expression of these enzymes ceased during the differentiation process (S1 Fig). We reasoned that this might be due to silencing of the CMV (cytomegalovirus) promoter at the (random) chromosomal integration locus of the expression cassette and decided to target the expression cassette to a 'safe harbor' locus in the mouse genome. Since two well established safe harbor loci (the Rosa26 and HPRT locus) are used in this cell line for doxycycline-controlled expression of the ASCL1 neurogenic transcription factor [21,23], we wanted to test if the mHipp11 intergenic region [29] can be targeted for transgenic expression of biotin ligases. We thus generated donor plasmids expressing a fusion protein of the MicroID biotin ligase [24], eGFP, and two copies of the aptamer-binding MS2 coat protein (MCP; [27,30]) either from a CMV or mPGK promoter. The integration cassette was designed to contain two mHipp11 homology arms that flank a unique SpeI site in the mHipp11 intergenic region (Fig 1A). To stably express the biotin ligase construct from the mHipp11 locus, we applied a CRISPR/Cas9 approach to deliver the donor vector, a recombinant Cas9, and chemically modified gRNA to mESCs via nucleofection. After selection with Hygromycin, monoclonal cell lines were generated by FACS sorting, followed by genotyping via PCR screening (Fig 1B). Of the five monoclonal cell lines screened for each construct, four were genotyped to contain the transgene successfully integrated, representing a high target efficiency.

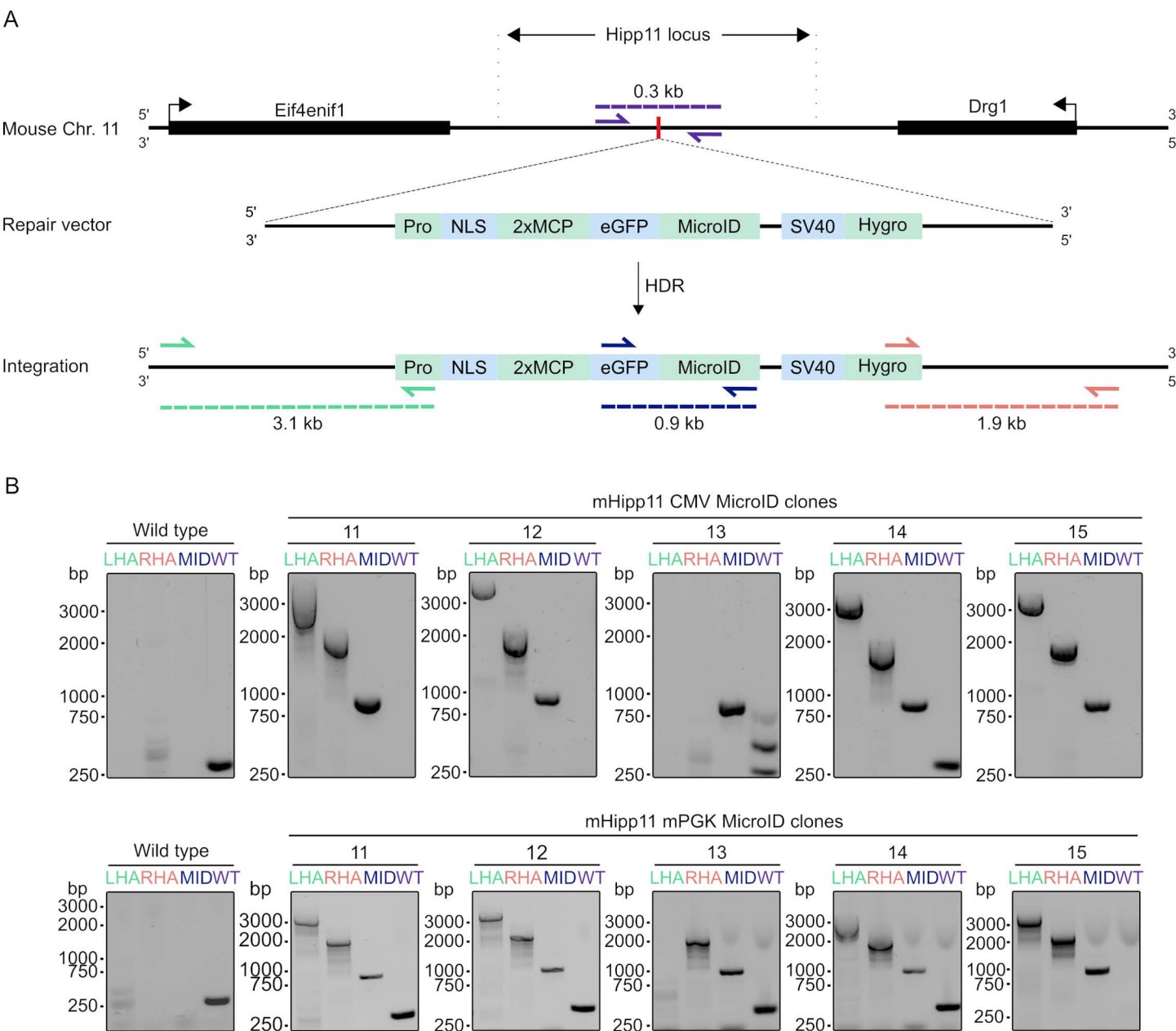

**Fig 1. CRISPR/Cas9 mediated integration of two different MicroID constructs into mHipp11 locus.** (A) The mouse mHipp11 locus is located between the Eif4enif1 and Drg1 genes on chromosome 11. CRISPR/Cas9 was used to create a double-stranded break at the unique SpeI site in the mHipp11 intergenic region (vertical red line). Via the homology-directed repair (HDR)-mediated repair pathway, a MicroID-containing expression construct (either under a CMV or mPGK promoter, 'pro') was stably integrated. (B) PCR detection for successful MicroID integration with primers for the left homology arm (LHA, green), right homology arm (RHA, orange), MicroID insert (MID, blue) and the WT allele (purple). The expected sizes for each PCR product are depicted in (A).

## Expression of a MicroID biotin ligase from the mHipp11 locus in mESCs

To confirm that the 2xMCP-eGFP-MicroID constructs are expressed in mESCs, we performed live cell imaging of cell lines with 2xMCP-eGFP-MicroID expressed from a mPGK (S2A Fig) or CMV promoter (S2B Fig), both stably integrated at a random genomic locus or at the mHipp11 locus. eGFP signal was detected in both cases, suggesting expression of

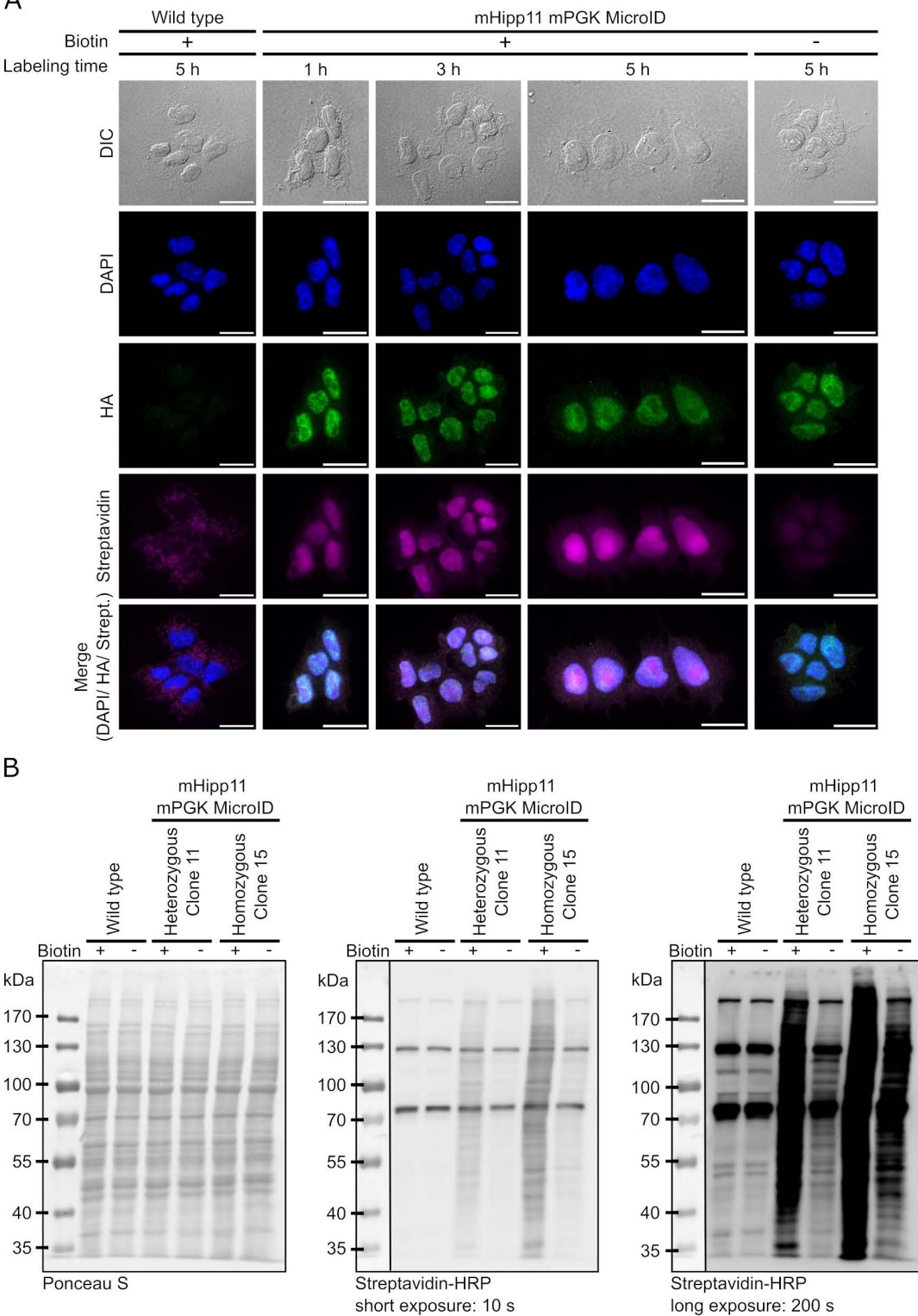

**Fig 2. MicroID is expressed and active in mESCs.** (A) Immunofluorescence images of wildtype and homozygous mHipp11 mPGK MicroID mESCs (clone 15). Nuclei were stained with DAPI, MicroID detected with an anti-HA antibody, and biotinylated proteins using streptavidin-Alexa Fluor 647. Bottom lane depicts merges of fluorescence images. Scale bars: 20 μm. (B) MicroID biotinylation activity was confirmed by western blot using a streptavidin-HRP conjugate for a heterozygous (clone 11) and a homozygous (clone 15) mHipp11 cell line expressing mPGK MicroID. Ponceau S staining served as loading control.

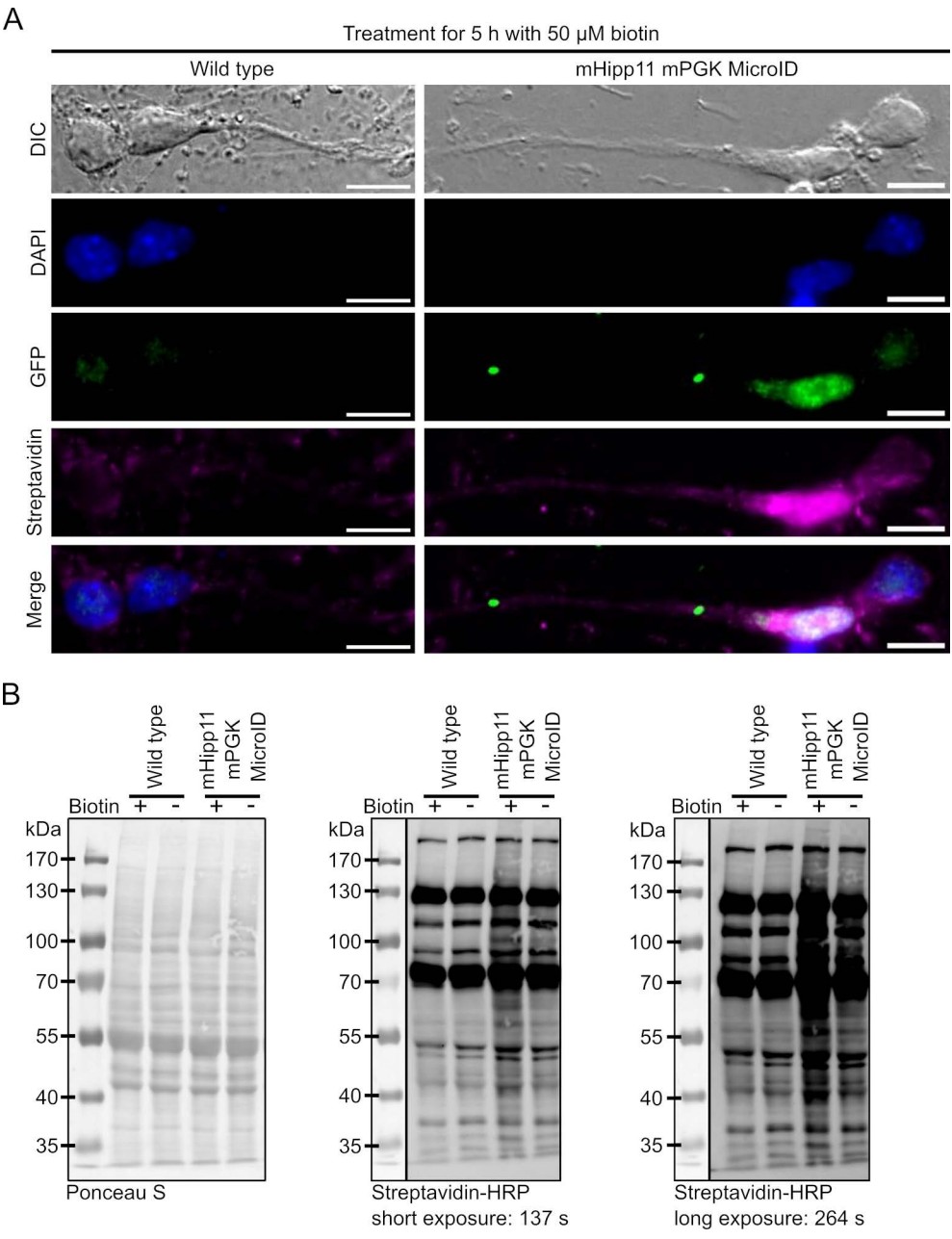

**Fig 3. MicroID is active in mESC-derived neurons when stably integrated into the mHipp11 gene locus and expressed from a mPGK promoter.** (A) Immunofluorescence staining of wildtype and mHipp11 mPGK MicroID mESC-derived neurons (clone 15). Nuclei were stained with DAPI, MicroID-GFP fusion protein detected with an anti-GFP antibody, and biotinylated proteins using streptavidin-Alexa Fluor 647. Bottom lane depicts merges of fluorescence images. Scale bars: 20 μm. (B) Biotinylation activity of a homozygous mPGK MicroID expressing cell line was confirmed by western blot analysis using streptavidin-HRP conjugate. Ponceau S served as loading control.

the fusion protein in mESCs independent of the integration site. Since the integrated biotin ligase construct is expressed, we next validated the biotinylation activity of MicroID when expressed from a CMV or mPGK promoter in heterozygous and homozygous clones (Fig 2 and S2C Fig). Under standard culturing conditions (without additional biotin in the medium) *in situ* staining of biotinylated proteins by streptavidin-coupled Alexa Fluor 647 (Fig 2A) and

western blot analysis using streptavidin-HRP (Fig 2B and S2C Fig) revealed modest levels of biotinylated proteins in MicroID expressing clones as compared with wild type mESCs. Addition of 50 μM biotin to tissue culture medium, however, resulted in a visible stimulation of biotinylation in MicroID clones (heterozygous clone 11 and homozygous clone 15) versus cells treated without additional biotin and wild type (Fig 2 and S2C Fig). Noteworthy, *in situ* detection of biotin in clone 15 revealed that the biotinylation pattern overlaps with the nuclear signal of the MicroID enzyme (Fig 2A), indicating that the increased signal detected by western blotting is due to local biotinylation. We conclude from these observations that the MicroID fusion protein is expressed in all mESC variants tested and that, in these cells, expression and function is independent of the promoter that drives expression of the construct.

## mHipp11 MicroID expressing cells display activity in mESC-derived neurons

We decided to continue all further experiments in mESCs and mESC-derived neurons with a homozygous line expressing MicroID at the mHipp11 locus from a mPGK1 promoter (clone 15) as previous reports have implicated that the CMV promoter is often silenced over time [31,32]. Differentiation into mESC-derived neurons was performed using an established protocol [22]. Loss of pluripotency was verified by staining for stem cell markers Oct3/4 and neuronal markers NF200 and MAP2 (S3 Fig). To ensure that our construct is expressed and active in mESC-derived neurons, we performed live-cell imaging, western blot analysis using a streptavidin-conjugate, and fluorescence microscopy using streptavidin-coupled Alexa Fluor 647 or Alexa Fluor 594 (Fig 3 and S4 Fig). Immmunofluorescence revealed that the 2xMCP-eGFP-MicroID construct is expressed and correctly located in the nucleus in mHipp11 mPGK MicroID mESC-derived neurons (Fig 3A). To assess biotinylation activity of the MicroID enzyme, mESC-derived neurons were cultured with 50 μM biotin and biotinylation assessed by in situ staining using Alexa Fluor647-coupled streptavidin (Fig 3A) or western blotting (Fig 3B). In contrast to wild type cells, addition of biotin to mHipp11 mPGK MicroID mESC-derived neurons resulted in stronger biotinylation (Fig 3). Furthermore, the biotinylation signal overlapped with the nuclear location of the MicroID fusion protein (Fig 3A).

## MicroID labeling in mESCs versus neurons

Having determined that the homozygous mHipp11 mPGK MicroID cell line (clone 15) allows expression and biotinylation of proteins in both mESCs and mESC-derived neurons, we next compared biotinylation efficiency in mESCs versus mESC-derived neurons. To do so, cell lines were cultured in the presence of 50 μM biotin and biotinylated proteins were detected by streptavidin-HRP western blotting (Fig 4A). This revealed that the MicroID is actively biotinylating proteins in both mESCs and mESC-derived neurons, although labeling was stronger in mESCs. To investigate if this is due to different expression levels, we performed RT-qPCR (Fig 4B). We confirmed that the biotin ligase construct is expressed from the mHipp11 safe-harbor locus. However, we also observed reduced expression levels in mESC-derived neurons compared to mESCs, which explains the observed reduced activity. A similar observation was made on the protein level. Since we failed to obtain MicroID fusion protein signals in these experiments in lysates of mESC-derived neurons, we performed GFP pulldown experiments from these lysates, followed by detection of the fusion protein by western blot. Although the fusion protein could be detected after enrichment by the pulldown, levels were much lower compared to those in mESCs (S5 Fig).

To rule out that our observations of the expression silencing in mESC-derived neurons are specific for the MicroID biotin ligase or mPGK promoter, we next tested the expression of 2xMCP-eGFP-TurboID fusion protein driven by a CMV promoter and randomly integrated into the mESC genome. Biotinylation activity in the corresponding cells before and during differentiation was compared to mESCs expressing the MicroID constructs after random integration or targeted to the mHipp11 locus. Consistent with our observation that 2xMCP-eGFP-MicroID expression is silenced upon differentiation (S1 Fig) when the fusion protein is expressed from a CMV promoter at a random site in the genome, we detect far less transgene-dependent biotinylation activity in mESC-derived neurons (Fig 4C, 'CMV-MicroID'). A similar observation is made with an expression construct of the same fusion protein driven from a mPGK promoter and integrated at a random location ('mPGK-MicroID') or with an expression construct of 2xMCP-eGFP-TurboID driven from a CMV promoter and integrated at a random location ('CMV-TurboID'). This confirms that the biotin ligase is more active in mESC-derived neurons when expressed from the mHipp11 safe-harbor locus compared to the version that is randomly integrated into the genome.

## Discussion

Proximity labeling approaches aimed to identify molecular partners of a bait protein or RNA *in vivo*. Although CRISPR/Cas can be used for direct genomic tagging of a bait protein with the sequence encoding a biotin ligase [33], these approaches often rely on the expression of the biotin ligase as a transgene. This is particularly the case when the enzyme is targeted to cellular compartments via fusion to localizing peptides [34,35] or sequestered to specific RNAs via fusion to RNA-binding domains [27,36]. Transgenic expression of proteins can be hampered by silencing, especially when transgenes are randomly integrated into the genome (see [5] for an overview). Since many reported proximity labeling approaches rely on transient transfection [37,38], silencing has not been discussed in the literature. However, a major drawback of transient transfection of biotin ligases is the potential for interexperimental variation due to uncontrolled overexpression of transgenes. Interestingly, in several cases, where transgenes were integrated into the genome, no silencing was reported [35]. This may be due to the use of immortalized tumor cell lines, which rarely show such gene silencing after random genomic integration [5], or the use of higly active, inducible promoters [24,39,40]. In previous work on targeting a biotin ligase (BirA*; [39,41,42]) to ß-actin mRNA we found that the BirA*-containing fusion protein must be integrated into the genome of mouse embryonic fibroblasts to achieve reliable expression [27]. Applying a similar integration strategy to mESCs and cells differentiated from these mESCs, it became apparent that random genomic integration of a similar expression construct rapidly results in silencing during differentiation (S1 Fig; Fig 4). Inactivation in mESC-derived neurons was independent of the promoter used for expression or the type of biotin ligase (Fig 4). To overcome silencing, we stably integrated and expressed a transgene encoding a biotin ligase from a 'safe harbor' locus. We chose the mHipp11 locus because two other well-studied loci, Rosa26 and HPRT, are already used in the ASCL1 mESC line used in our experimental setup [21,23,43]. The mHipp11 locus has already been successfully used to express other transgenes such as human CD1A in murine macrophages and dendritic cells [44] or TIR1 in mESCs, hESCs and differentiated B6^V1-TIR1 mESCs [45]. Integration at the mHipp11 locus has also been shown to result in expression in mESCs as well as in neuronal cells derived from them [45]. Consistent with these results, we found that a fusion protein of eGFP, the biotin ligase MicroID [24], and the MS2 coat protein (MCP; [27,30]) is expressed, correctly targeted to the nucleus, and functional as judged by its biotinylation activity (Fig 2 and 3). These results demonstrate that the Hipp11 locus can be

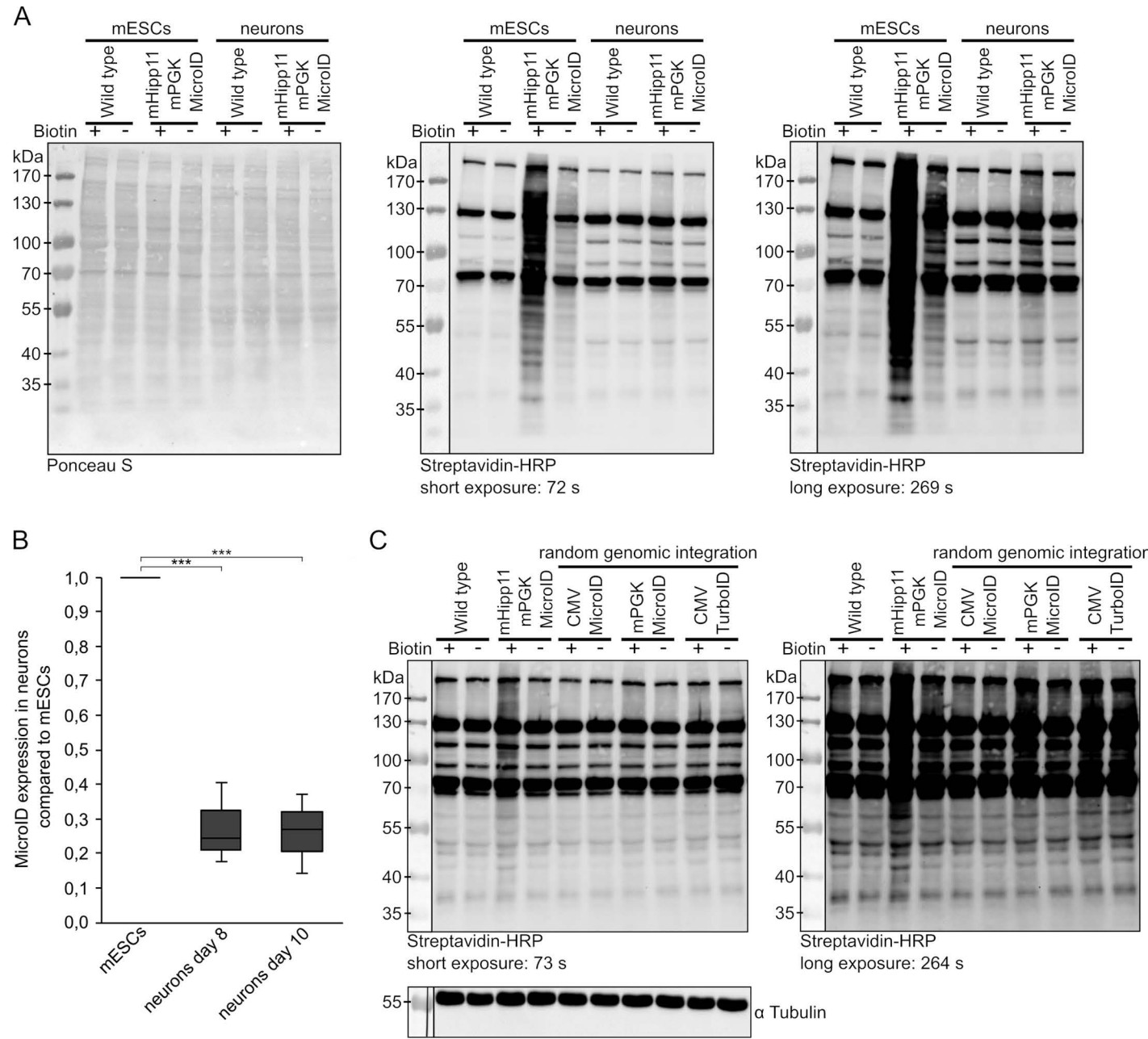

**Fig 4. Biotinylation activity in mESCs and mESC-derived neuronal cell lines.** (A) MicroID activity in mESCs vs. derived neurons. Cell lines were treated with 50 μM excess biotin for 5 h. Middle and right image show the same blot at different exposures. (B) RT-qPCR for MicroID expression in mHipp11 mPGK MicroID mESC-derived neurons normalized to mHipp11 mPGK MicroID mESCs and GAPDH. Results from three independent experiments are shown. Statistical significance was analyzed using an unpaired Student's t test. ***, p < 0.001. (C) Comparing biotin ligase activity of mESC-derived neuronal cell lines expressing either mPGK-MicroID from mHipp11 gene locus with mPGK- or CMV-MicroID, and CMV-TurboID after random integration into the genome. Left and right images show the same blot at different exposures. After stripping, the blot was reprobed with an anti-tubulin antibody to control for equal loading.

used for the stable expression of biotin ligases or fusion proteins containing them. However, we also observed that the activity of the MicroID enzyme was lower in mESC-derived neurons compared to undifferentiated mESCs. This was likely due to lower expression levels (Fig 4), suggesting either that the transgene is partially silenced or that the promoter we used (mPGK)

has a lower activity in neurons than in mESCs. A similar partial downregulation has already been observed for the TIR1 protein in B6$^{V1-TIR1}$ mESC-derived neurons [45]. Hoever, it should be noted, that in our ASCL1-inducible neurons, biotinylation activity of the MicroID enzyme is still observed in neuronal cells after 8 days of differentiation, even under constitutive expression conditions.

Our data thus demonstrate the potential of the mHipp11 gene locus for transgene targeting in cell lines with other commonly used safe harbor loci such as Rosa26 or HPRT already occupied. We also show that all three loci can be used in combination. This may be very useful for biotin ligases that cannot be expressed as a fusion protein from the corresponding endogenous locus and require their own expression elements. It will be particularly useful in BioID approaches aimed at characterizing of changing interactome patterns between non-differentiated cells and differentiating cells at different stages.

## Supporting information

**S1 Fig. Live cell imaging of 2xMCP-eGFP-MicroID or 2xMCP-eGFP-TurboID expressed in mESCs vs mESC-derived neurons.** The expression construct was randomly integrated into the genome. Only background signal is detected in neurons, whereas a strong nuclear eGFP signal in mESCs is present. Scale bars: 20 μm.
(TIFF)

**S2 Fig. MicroID-containing fusion protein is expressed from mPGK and CMV promoters and active in mESCs.** (A) Live imaging of 2xMCP-eGFP-MicroID expressed from a mPGK promoter after random integration into the mESC genome (second column) or targeted integration into the mHipp11 locus (four different clones shown). (B) Live imaging of 2xMCP-eGFP-MicroID expressed from a CMV promoter after random integration into the mESC genome (second column) or targeted integration into the mHipp11 locus (four different clones shown). (C) Biotinylation activity of two clones expressing 2xMCP-eGFP-MicroID from the mHipp11 locus. Biotinylated proteins are detected via streptavidin-HRP. Two different exposures of the same blot are shown. Strong biotinylation is only detectable in MicroID clones after addition of excess biotin. Scale bar: 20 μm.
(TIFF)

**S3 Fig. Immunofluorescence staining of stem cell pluripotency marker Oct3/4 and neuronal markers NF200 and Map2 in mESCs and mESC-derived neurons.** (A) Wild type and mHipp11 mPGK MicroID mESCs were stained with an anti-Oct3/4 antibody to validate pluripotency. Merged images show overlapping DAPI and Oct3/4 signal in the nucleus. Scale bars: 10 μm. (B) Wild type and mHipp11 mPGK MicroID mESC-derived neurons (clone 15) were differentiated until day 8 and day 10 and stained with anti-NF200 antibody and anti-MAP2 antibody to validate neuronal markers. Merged images are composites of DAPI, NF200 and MAP2 staining. Scale bar: 20 μm. (C) Wild type and mHipp11 mPGK MicroID mESC-derived neurons were stained with an anti-Oct3/4 antibody to verify loss of pluripotency after differentiation.
(TIFF)

**S4 Fig. MicroID is active in mESC-derived neurons when stably integrated into the mHipp11 gene locus and expressed from a mPGK promoter.** (A) MicroID construct expression in mESC-derived neurons was confirmed by live-cell imaging. Wild type cells show no nuclear eGFP signal. (B) Biotinylation activity in mESC-derived neurons at different time points was visualized by streptavidin-coupled Alexa Fluor 594. Scale bars: 20 μm.
(TIFF)

**S5 Fig. Streptavidin pulldown of biotinylated proteins in mESCs and mESC-derived neurons.** MicroID fusion protein (70 kDa, arrow) was detected by an anti-GFP antibody in total cell lysates (left lanes) and after enrichment by pulldown (right lanes).
(TIFF)

**S6 Fig. Raw data of gels and western blots displayed in Figures 1, 2, 4, S2 Fig, and S5 Fig.** .
(PDF)

**S1 Table. Primers used in this study.**
(PDF)

**S2 Table. Antibodies used in this study.**
(PDF)

## Acknowledgement

We would like to thank Vincent Kelly (Trinity College Dublin) for providing us with the mHipp11 knock-in vector, Marina Chekulaeva (Berlin Institute for Medical Systems Biology of the Max Delbrück Center) for the ASCL1 murine cell line, Julien Bethune (Hamburg University of Applied Science) for the plasmid carrying MicroID, Stefan Hauser (Hertie Institute for Clinical Brain Research Tübingen) for the MAP2 antibody, and Iliana Nikolou for help with RNA isolation.

## Author contributions

**Conceptualization:** Lisa Feicht, Ralf-Peter Jansen.

**Funding acquisition:** Ralf-Peter Jansen.

**Investigation:** Lisa Feicht, Aaron Dangel.

**Methodology:** Aaron Dangel.

**Project administration:** Ralf-Peter Jansen.

**Validation:** Lisa Feicht.

**Visualization:** Lisa Feicht.

**Writing – original draft:** Lisa Feicht, Aaron Dangel, Ralf-Peter Jansen.

**Writing – review & editing:** Lisa Feicht, Ralf-Peter Jansen.

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
