## [Decision Letter · Decision Letter 0]

7 Aug 2024

PONE-D-24-27880Expression of transgenic biotin ligases in inducible neuronal murine cell lines by integration into the mHipp11 gene locusPLOS ONE

Dear Dr. Jansen,

Thank you for submitting your manuscript to PLOS ONE. After careful consideration, we feel that it has merit but does not fully meet PLOS ONE’s publication criteria as it currently stands. Therefore, we invite you to submit a revised version of the manuscript that addresses the points raised during the review process. **Please carefully read the comments below from both reviewers and completely address their concerns.**

We look forward to receiving your revised manuscript.

Kind regards,

Chunming Liu

Academic Editor

PLOS ONE

Journal Requirements:

RPJ was funded by a grant of the Deutsche Forschungsgemeinschaft (grant no. DFG JA696/10-2) in the course of the DFG-funded Research Unit FOR2333 (https://gepris.dfg.de/gepris/projekt/270067186).

LF was supported by the International Max Planck Research School “From Molecules to Organisms” (https://www.phd.tuebingen.mpg.de/imprs).

3. We notice that your supplementary tables are included in the manuscript file. Please remove them and upload them with the file type 'Supporting Information'. Please ensure that each Supporting Information file has a legend listed in the manuscript after the references list.

Reviewers' comments:

Reviewer's Responses to Questions

**Comments to the Author**

1. Is the manuscript technically sound, and do the data support the conclusions?

Reviewer #1: Yes

Reviewer #2: Partly

2. Has the statistical analysis been performed appropriately and rigorously? 

Reviewer #1: N/A

Reviewer #2: No

3. Have the authors made all data underlying the findings in their manuscript fully available?

Reviewer #1: Yes

Reviewer #2: Yes

4. Is the manuscript presented in an intelligible fashion and written in standard English?

Reviewer #1: Yes

Reviewer #2: Yes

5. Review Comments to the Author

**Reviewer #1: ** Originality and Relevance: The study addresses a significant problem in the field of genetic engineering and cellular biology by exploring a method to ensure consistent expression of transgenes during cell differentiation. This is relevant for researchers working on gene expression in stem cells and differentiation studies.

Technical Quality and Rigorous Methodology: The methodology is robust, employing multiple experimental approaches to validate findings. The use of RT-qPCR, Western blotting, and immunofluorescence, biotinylation assay provide a comprehensive validation of results.

Interpretation and Validity: The interpretation of results is logical, and the conclusions are supported by the data presented. The study provides clear evidence that targeting the mHipp11 locus can minimize transgene silencing after differentiation.

Based on the evaluation, I recommend accepting the publication with minor revisions. The study is well-conducted and addresses an important issue in the field, providing valuable insights and a potential solution for transgene silencing during cell differentiation. However, the following minor revisions are suggested to enhance the clarity of the manuscript.

Please identify the clones used for subsequent experiments and please label them accordingly in both the figures, figure legends and the text. As of current, I can only see Clone 11 and Clone 15 data mentioned in Figure 2B. I request the authors to identify to the clones used in the subsequent Figures 2C, 3, 4.

Please also comment on the transgene silencing effects comparing Clone 11 and Clone 15, ideally providing data, if available.

Concern: The DAPI channel exposure for figure 2C (3h) is noticeably higher than the rest.

**Reviewer #2:**  In this manuscript, Feicht et al. showed that the Hipp11 locus can be utilized in combination with other transgenes located in other safe harbors. The authors present results showing that the Hipp11 safe harbor is a more reliable and less silenced locus than in the random integrations strategy. Although most of the data appears convincing, there are still a few major and minor comments that have to be addressed by the authors to strengthen the overall manuscript.

Major comments:

- The authors presented qPCR data of the MicroID expression in Figure 4B. However, the total protein level of MicroID fusion is still missing for mESC, Neuron Day 8, and Neuron Day 10 cells. Also, did the authors test the levels of MicroID longer than ten days to confirm (or not) that the silencing is not happening later?

- Similarly, it will be critical to evaluate the levels and subcellular localization of MicroID in mESC and neurons. This will help confirm the biotin labeling as well. Showing GFP images in live cells might show unwanted autofluorescence and background. Immunostaining the GFP fusion can help enhance the specific signal and lessen the non-specific background, especially in a non-GFP channel.

- The authors will have to validate the pluripotency markers of the clone(s) used in Figure 3. After differentiation, neuronal markers have to be used to validate the derived neurons.

- In Figure 3C, right panel, the images in red are not convincing. They will need to be shown with higher resolution, ideally co-stained with the MicroID fusion protein.

- In Figure 3A, the authors stated that the wild-type (WT) cells show no nuclear staining. Fig 3A lacks DNA staining to be able to conclude with confidence the exact subcellular localization. Also, WT cells already display GFP-positive accumulation likely in the cell body. These issues will have to be addressed.

- Conclusions in lines 334–342: The authors will have to re-formulate these conclusions since we still see a very faint biotinylation effect coming from potential active biotin ligase in + biotin conditions in random genomic integration for mPGK MicroID and CMV TurboID (Fig4C).

Minor Comments:

- We do not see in the result section which clone(s) the authors followed up with in Figure 3.

- The authors will have to perform statistical analysis for fig4B and quantify the levels of AF594 for fig3C (or the updated fig3C).

6. PLOS authors have the option to publish the peer review history of their article (what does this mean? ). If published, this will include your full peer review and any attached files.

**Do you want your identity to be public for this peer review?** For information about this choice, including consent withdrawal, please see our Privacy Policy .

Reviewer #1: **Yes: ** Arthur Luhur

Reviewer #2: No

---

## [Author Response · Author response to Decision Letter 1]

17 Oct 2024

Our response to the reviewer's comments can be found in the attached text file "Response to Reviewers".

---

## [Decision Letter · Decision Letter 1]

13 Nov 2024

PONE-D-24-27880R1Expression of transgenic biotin ligases in inducible neuronal murine cell lines by integration into the mHipp11 gene locusPLOS ONE

Dear Dr. Ralf-Peter Jansen,

Thank you for submitting your manuscript to PLOS ONE. After careful consideration, we feel that it has merit but does not fully meet PLOS ONE’s publication criteria as it currently stands. Therefore, we invite you to submit a revised version of the manuscript that addresses the points raised by Reviewer #2.

We look forward to receiving your revised manuscript.

Kind regards,

Chunming Liu

Academic Editor

PLOS ONE

Journal Requirements:

Reviewers' comments:

Reviewer's Responses to Questions

**Comments to the Author**

1. If the authors have adequately addressed your comments raised in a previous round of review and you feel that this manuscript is now acceptable for publication, you may indicate that here to bypass the “Comments to the Author” section, enter your conflict of interest statement in the “Confidential to Editor” section, and submit your "Accept" recommendation.

Reviewer #1: All comments have been addressed

Reviewer #2: (No Response)

2. Is the manuscript technically sound, and do the data support the conclusions?

Reviewer #1: Yes

Reviewer #2: Yes

3. Has the statistical analysis been performed appropriately and rigorously? 

Reviewer #1: Yes

Reviewer #2: Yes

4. Have the authors made all data underlying the findings in their manuscript fully available?

Reviewer #1: Yes

Reviewer #2: Yes

5. Is the manuscript presented in an intelligible fashion and written in standard English?

Reviewer #1: Yes

Reviewer #2: Yes

6. Review Comments to the Author

Reviewer #1: All comments have been addressed. Both figure and text revisions have been completed as requested and this has improved the clarity and quality of report.

Reviewer #2: The authors addressed most of the comments. Below are a few minor follow-ups to finalize this review.

Minor Comments:

- The S5 supplementary figure shows red saturated bands. Unless this is required by the journal, these blots are better shown in grayscale.

- The S6 fig is not the best quality wise; it would be still interesting and important to show this data as a supplementary figure and briefly touch on this challenging aspect in the text, so the scientific community is aware of such information.

7. PLOS authors have the option to publish the peer review history of their article (what does this mean? ). If published, this will include your full peer review and any attached files.

**Do you want your identity to be public for this peer review?** For information about this choice, including consent withdrawal, please see our Privacy Policy .

Reviewer #1: No

Reviewer #2: No

---

## [Author Response · Author response to Decision Letter 1]

23 Nov 2024

Minor comments from reviewer 2 have been addressed and suggestions of reviewer 2 for additions / corrections have been made accordingly.

---

## [Decision Letter · Decision Letter 2]

3 Dec 2024

Expression of transgenic biotin ligases in inducible neuronal murine cell lines by integration into the mHipp11 gene locus

PONE-D-24-27880R2

Dear Dr. Ralf-Peter Jansen,

We’re pleased to inform you that your manuscript has been judged scientifically suitable for publication and will be formally accepted for publication once it meets all outstanding technical requirements.

Kind regards,

Chunming Liu

Academic Editor

PLOS ONE

Additional Editor Comments (optional):

Reviewers' comments:

Reviewer's Responses to Questions

**Comments to the Author**

1. If the authors have adequately addressed your comments raised in a previous round of review and you feel that this manuscript is now acceptable for publication, you may indicate that here to bypass the “Comments to the Author” section, enter your conflict of interest statement in the “Confidential to Editor” section, and submit your "Accept" recommendation.

Reviewer #2: All comments have been addressed

2. Is the manuscript technically sound, and do the data support the conclusions?

Reviewer #2: Yes

3. Has the statistical analysis been performed appropriately and rigorously? 

Reviewer #2: Yes

4. Have the authors made all data underlying the findings in their manuscript fully available?

Reviewer #2: Yes

5. Is the manuscript presented in an intelligible fashion and written in standard English?

Reviewer #2: Yes

6. Review Comments to the Author

Reviewer #2: (No Response)

7. PLOS authors have the option to publish the peer review history of their article (what does this mean? ). If published, this will include your full peer review and any attached files.

**Do you want your identity to be public for this peer review?** For information about this choice, including consent withdrawal, please see our Privacy Policy .

Reviewer #2: No

---

## [Editor Report · Acceptance letter]

PONE-D-24-27880R2

PLOS ONE

Dear Dr. Jansen,

I'm pleased to inform you that your manuscript has been deemed suitable for publication in PLOS ONE. Congratulations! Your manuscript is now being handed over to our production team.

Kind regards,

on behalf of

Dr. Chunming Liu

Academic Editor

PLOS ONE